



# A Bayesian data assimilation framework for lake 3D hydrodynamic models with a physics-preserving particle filtering method using `SPUX-MITgcm` v1

Artur Safin[1], Damien Bouffard[1], Firat Ozdemir[2], Cintia L Ramón[3,1], James Runnalls[1], Fotis Georgatos[2], Camille Minaudo[4], and Jonas Šukys[1]

[1]Eawag: Swiss Federal Institute for Aquatic Science and Technology, Switzerland
[2]Swiss Data Science Center, Switzerland
[3]Water Research Institute and Department of Civil Engineering, University of Granada, Spain
[4]École Polytechnique Fédérale de Lausanne, Switzerland

**Correspondence:** Artur Safin (artur.safin@eawag.ch)

**Abstract.** We present a Bayesian inference for a three-dimensional hydrodynamic model of Lake Geneva with stochastic weather forcing and high-frequency observational datasets. This is achieved by coupling a Bayesian inference package, `SPUX`, with a hydrodynamics package, `MITgcm`, into a single framework, `SPUX-MITgcm`. To mitigate uncertainty in the atmospheric forcing, we use a smoothed particle Markov chain Monte Carlo method, where the intermediate model state posteriors are resampled in accordance with their respective observational likelihoods. To improve the assimilation of remotely sensed temperature, we develop a bi-directional Long Short-Term Memory (Bi-LSTM) neural network to estimate lake skin temperature from a history of hydrodynamic bulk temperature predictions and atmospheric data. This study analyzes the benefit and costs of such state of the art computationally expensive calibration and assimilation method for lakes.

## 1 Introduction

Lake management is a constantly evolving trade-off between different conflict of use. The most obvious is that lakes are easily accessible sources of drinking water but also the place where wastewater is ultimately discharged. Lake stakeholders traditionally evaluate the evolution of lakes from in situ observations. While still not widely used for this purpose, previous studies clearly showed the benefit of one and three dimensional hydrodynamic models to project different scenarios for the short or long-term future (Gaudard et al., 2019; Soulignac et al., 2019; Vinnå et al., 2021).

While a number of dedicated monitoring projects already exist for a number of large lakes, operational fully three-dimensional (3D) models are yet quite sparse. The most notable is the NOAA Great Lakes Operational Forecast System (GLOFS) (Chu et al., 2011; Anderson et al., 2018), which provide comprehensive predictions (water temperature, velocity and level, and ice cover) for all the Laurentian Great Lakes. Over 25 years, the forecasting service has been continuously improved with better and more sophisticated models. Currently, data assimilation is used for calibration only (Anderson et al., 2018), but there is research toward making it part of the operational mode as well (Ye et al., 2020). Another platform is Meteolakes, which provides short-term water temperature and velocity forecasts for Lakes Geneva, Biel, Zurich and Greifen in Switzerland (Baracchini,





2019; Baracchini et al., 2020a, b). The platform uses an ensemble Kalman filter to assimilate remotely sensed lake surface water temperature (LSWT), which reduced the mean temperature prediction error by half. An additional benefit of the ensemble filter was a better prediction of mesoscale physical processes such as gyres and upwellings (Baracchini et al., 2020a). However,

due to the limitation of the assimilation scheme, only a fraction ($\approx$3.7%) of the available LSWT images were used.

The purpose of this study is to investigate a novel approach to data assimilation of highly heterogeneous data using Bayesian inference techniques applied to a 3D hydrodynamic model of Lake Geneva. The model relies on the ensemble affine invariant sampler (EMCEE) (Goodman and Weare, 2010; Šukys and Bacci, 2021) to calibrate distributions of physical model parameters. The advantage of this approach over standard inference methods is that it provides a more informative and accurate parameter

estimation, albeit at higher computational expense. To increase the confidence in the sampling algorithm, we used a particle method that provides trajectories consistent with the hydrodynamic model (Andrieu et al., 2010; Šukys and Bacci, 2021). In particular, as a substitute to the more well-known Kalman Filter and the 4D-Var algorithms, the trajectories themselves are resampled based on their respective observational likelihoods, with the more probable realizations stochastically forming the basis for sequential predictions.

To facilitate the assimilation of remotely sensed lake surface water temperature (LSWT), we deploy a Bi-directional Long Short-Term Memory (Bi-LSTM) neural network to estimate the skin temperature of the lake and to quantify its uncertainty. The network relies on a 27-hour history of hydrodynamic model bulk temperature and atmospheric predictions as inputs for the conversion. The neural network was trained using 28 months of data (2018, 2020 and Jan-Apr 2021) using MeteoSwiss COSMO-1 atmospheric model reanalysis and Meteolakes water bulk temperature predictions. We obtained a 33% root-mean

square error (RMSE) reduction with more precise uncertainty estimates in comparison to the model presented in Lieberherr and Wunderle (2018).

We present the openly available `SPUX-MITgcm` framework, which integrates the Bayesian inference algorithms of the `SPUX` package (Šukys and Bacci, 2021) with the hydrodynamics of the `MITgcm` code (Adcroft et al., 1997) and the trained BiLSTM network. To the best of our knowledge, the data assimilation and particle filtering approach that we propose in this

paper have not been previously tested for fully three-dimensional models due to the relatively high computational costs of model parameter calibration and lack of supporting software. The results of this framework demonstrate the viability of this approach and serve as a proof of concept for other higher-dimensional problems.

## 2   Data and Numerical Model

In this section, we describe the available data, the hydrodynamic model and our data assimilation approach. As data and

software reproducibility are essential to more open and accessible research, in the supplementary material, we provide documentation on accessing and running the numerical model, which enables a full replication of the results over a short period of time we present in this paper.



**Table 1.** Hydrodynamic model parameters.

| Parameter | Value | Units |
|---|---|---|
| Quadratic bottom drag coefficient $C_D$ | 0.0025 | none |
| Constant salinity value | 0.05 | psu |
| Coriolis parameter $f_0$ | $1.068 \cdot 10^{-4}$ | 1/s |
| Background vertical viscosity | $10^{-6}$ | m$^2$/s |
| Background vertical diffusivity | $1.4 \cdot 10^{-7}$ | m$^2$/s |
| Adams-Bashforth | 0.03 | none |
| Non-dimensional lateral eddy viscosity | $6 \cdot 10^{-4}$ | none |
| Lateral eddy diffusivity | 0.5 | m$^2$/s |

## 2.1 Study site

Lake Geneva is the largest freshwater lake in Western Europe located on the border between France and Switzerland covering
an area of approximately 580 km$^2$ with an average depth of 154 m. Spanning 73 km along its longest axis and with a maximum
width of 14 km, the lake consists of a wider and deeper main portion in the east and a narrow and shallow portion in the west.
The water level and discharge rate into the Rhône river are managed by a dam on the western end of the lake. Lake Geneva is
predominantly vertically stratified in density due to temperature, although complete mixing does occur every few years. The
mountainous nature of the region significantly affects the wind patterns over the lake, with north-east and south-west being
the prevalent directions. These wind patterns, along with seasonal variability in light penetration depth, significantly affect the
thermal structure of the lake (Bouffard and Lemmin, 2013; Bouffard et al., 2018). As the mean water residence time in the lake
is 10 years, the primary factor driving the lakes' dynamics is the atmospheric forcing.

## 2.2 Hydrodynamic model

We simulate the hydrodynamics of Lake Geneva using the MITgcm (Adcroft et al., 1997) package (tag 'checkpoint67q'), which
uses the finite volume method to solve the incompressible Navier-Stokes equations under the Boussinesq approximation. Alter-
native packages were FVCOM (Chen et al., 2006) used by GLERL, and Delft3D-FLOW (Deltares, 2013) used by Meteolakes.
We use a hydrostatic formulation combined with a third-order direct space-time flux limiter advection scheme (Prather, 1986).
A nonlinear equation of state by McDougall et al. (2003) is applied with constant salinity. Due to the large size of the lake, the
Coriolis force is included. A detailed list of fixed model parameter values is provided in Table 1.

The simulations are performed on a Cartesian grid ($z$-coordinate system) with 1 km horizontal resolution and 50 vertical
layers that gradually increase in thickness from 1 meter at the surface to 21 meters in the deepest portion of the lake. We chose
a timestep of 60 seconds. While larger timesteps were still numerically stable, a smaller value was helpful in reducing vertical





temperature over-diffusion into the deep layers. To improve the accuracy of the topography and reduce spurious artifacts near the bottom, "shaved" cells are allowed. We apply free-slip boundary conditions to both horizontal and vertical boundaries and

use non-dimensional bottom drag coefficient from Bouffard and Lemmin (2013) to enable energy dissipation. On the surface, we use an implicit free surface formulation.

We model the vertical mixing processes using the nonlocal K-Profile Parameterization (KPP) scheme (Large et al., 1994), which is commonly used in oceanography. Small background vertical diffusivity and viscosity parameters are included to ensure stability (see Table 1). For the equivalent parameters on the lateral scales, we manually tuned eddy viscosity and

diffusivity parameters by minimizing the difference between model predictions and in situ temperature profiles at Buchillon station. While a more optimal approach is to infer these parameters, our data assimilation scheme found them difficult to identify (see supplementary material).

Surface forcing inputs were derived from the MeteoSwiss COSMO-E numeric weather prediction model which are made at 2.2 km resolution. While the COSMO-E model generates an ensemble of 21 predictions, in previous hydrodynamic mod-

els of Lake Geneva only the mean and spread were used (Baracchini et al., 2020b; Cimatoribus et al., 2018). In Sect. 2.4.2, we describe a data assimilation approach that makes use of the individual ensembles, which represent the span of weather dynamics more accurately. This approach has the additional advantage of not requiring the estimation of the spatio-temporal noise parameters that Baracchini et al. (2020b) used to add stochasticity to their model. Air pressure, air temperature, wind velocity, longwave radiation, relative humidity and cloud coverage are used to determine the input fields in accordance with

Fink et al. (2014), where the wind-drag coefficients of Wüest and Lorke (2003) are used to improve surface stress coupling at low wind speeds. We also include the inflow and outflow of the Rhône river using the volume flow and temperature measured a few kilometers upstream at Porte du Scex (Station Federal Office for Environment, FOEN). As smaller tributaries and precipitation/evaporation are not taken into consideration in the model, the water level in the model is manually adjusted to the measured values from the St. Prex station.

Correct transfer of heat and energy from the atmosphere is an essential component of a well-performing hydrodynamic model, especially in the summer. In this regard, the bulk transfer coefficient of sensible heat (Dalton number) is a significant parameter that in several studies (Verburg and Antenucci, 2010; Baracchini, 2019; Rahaghi et al., 2018) has been shown to be larger than the default values used in ocean simulations. Therefore, we seek to infer this parameter. In addition, to more realistically accommodate fluctuations in water transparency, we estimate a spatially uniform Secchi depth using the

photosynthetically active radiation (PAR) data from the LéXPLORE moorings (Wüest et al., 2021) to determine the attenuation rate of shortwave energy in the water column. In the future, given the spatial variability of Lake Geneva (Bouffard et al., 2018; Soulignac et al., 2019), a better approach might be to use remote sensing data to estimate Secchi depth for different lake locations. Finally, we use the albedo formula of Cogley (1979) to account for seasonal changes in surface reflectivity. The Secchi measurements and albedo values are visualized in the supplementary material.





**Table 2.** Characteristics of the in situ datasets. Note that we assume the FOEN river and water level data to be exact.

| Dataset | Physical quantity | Frequency used in the model | Depth span (m) | Sensor count | Sensor uncertainty |
|---------|-------------------|-----------------------------|----------------|--------------|--------------------|
| Buchillon | Temperature | hourly | 1, 35 | 2 | 0.1°C |
| LéXPLORE | Temperature | hourly | 0.25–90 | 16 | 0.1°C |
| LéXPLORE | Velocity magnitude | hourly | 15–90 | 8 | 0.08 m/s |
| GE3 | Temperature | 1-2 measurements per month | 2.5–50 | 8 | 0.1°C |
| SHL2 | Temperature | 1-2 measurements per month | 2.5–290 | 16 | 0.1°C |
| FOEN Rhone inflow/outflow | Temperature, volume flow | hourly | - | 2 | - |
| FOEN St.Prex | Water Level | hourly | surface | 1 | - |
| LéXPLORE | PAR | daily | 0-30m | 4 | - |

## 2.3 Observational datasets

A particular advantage of the Bayesian framework is the natural ability to handle multiple sources of data with their respective uncertainties. For Lake Geneva, the observations are either in the form of an in situ measurement or remotely sensed surface temperature.

### 2.3.1 In situ

In situ datasets used in the simulations are summarized in Table 2 and their locations are displayed in Fig. 1. To make the data assimilation process much more manageable, the data from LéXPLORE and BAFU have been subsampled to the hourly rate from the original intervals of 5 seconds and 10 minutes, respectively. The vertical resolution of the LéXPLORE dataset was also reduced to match the model discretization levels. Finally, due to the coarse horizontal resolution of the model, only the magnitude of velocity was considered as means of calibrating the kinetic energy of the lake.

### 2.3.2 Remotely sensed temperature

A processing chain from the University of Bern enables the extraction LSWT images at a resolution of 1 km from the orbital Advanced Very High Resolution Radiometer (AVHRR). On average, typically 10 images of Lake Geneva are generated per day, of which around 2 are deemed usable by the retrieval process. The quality of an individual snapshot is affected by a number of factors, such as the zenith angle, cloudiness or sensor errors (Riffler et al., 2015; Kilpatrick et al., 2001). In Lieberherr and Wunderle (2018), a system of assigning a quality flag (QF) for the different satellite measurement conditions was developed. An analysis based on in situ lake data provided an estimate of the uncertainties and biases, ranging from 1.3 °C for QF 6 to 1.5 °C for QF 1. However, we believe that a more accurate uncertainty model can be established, and therefore in Sect. 2.4.3 we detail an alternative approach based on machine learning that uses a history of model predictions and weather conditions to generate a bulk-to-skin estimate together with the associated uncertainty.



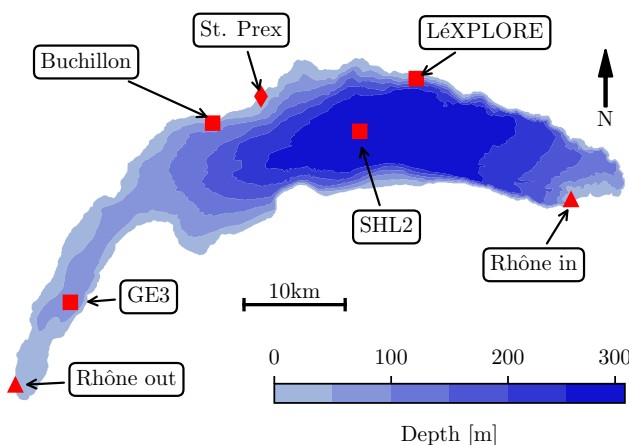

**Figure 1.** Location of the measurement sensors on Lake Geneva. The plot also shows the Rhone inflow and outflow sensor locations, as well as the St. Prex station, which measures the water level.

## 2.4 Data assimilation

Lakes, akin to the atmosphere, are highly volatile and sensitive systems. For hydrodynamic models, this means that small perturbations in state variables can significantly affect the resulting model trajectory. Due to the multiple sources of uncertainty present in numerical discretization models, most importantly uncertainty in forcing terms, such trajectory deviations are ultimately unavoidable. The remedy comes in the form of *data assimilation* (DA), a framework for providing trajectory corrections based on actual observations.

### 2.4.1 Choice of DA model

In meteorological and aquatic forecasting models, DA is an essential tool for improving predictive performance (Lahoz and Schneider, 2014; Ye et al., 2020; Baracchini et al., 2020b). A variety of DA algorithms have been proposed and deployed in operational setting, and offer different performance-to-optimality trade-offs. The two primary subfields are variational DA and sequential methods (Lahoz and Schneider, 2014). In the variational approach, an objective function describing the discrepancy between the model and observations is sought to be minimized. 4D-Var is the most popular form and offers the capability to transfer information from observed regions to unobserved for non-linear models. Some of the drawbacks of this approach are the relatively large numerical costs both for the model and uncertainty, reduced flexibility, and complex implementation of time-dependent parameters (Lahoz and Schneider, 2014; Baracchini et al., 2020b). From the sequential methods, Kalman filters (KF), which iteratively evolve a forecast together with error covariance matrices, are an optimal approach. As the true KF is expensive due to the cost of computing the covariance matrix in model space, local model linearity is frequently assumed. For high-dimensional systems, this however can result in significant errors in the state and covariance estimation.





Significant performance enhancement is enabled through ensemble methods, where a collection of model states are propagated and the resulting trajectories enable the estimation of covariance. Ensemble data assimilation is particularly attractive, as

it significantly relaxes the requirements from the model. The ensemble KF (EnKF) is an efficient and highly popular blend, providing greater stability and easier covariance estimation. The EnKF has successfully been applied to hydrodynamic forecasting of Lake Geneva by Baracchini et al. (2020a), with a 54% reduction in temperature error in comparison to an unassimilated model. In recent years, the local ensemble transform KF (LETKF) has been tested in a number of weather prediction frameworks with encouraging results (Gustafsson et al., 2018).

The assimilation techniques introduced above have the limitation of assuming that uncertainties and model states are Gaussian and the model is linear (Lahoz and Schneider, 2014). While this assumption is perfectly reasonable for many applications, non-Gaussian observational error and parameter distributions can be problematic for such methods. For higher resolution models, the traditional approaches of 4D-Var and EnKF have shown declining performance at convective scale (Gustafsson et al., 2018; van Leeuwen et al., 2019). To mitigate these challenges, particle filter (PF) methods have been proposed, which allow

non-Gaussian distributions and are particularly suitable for non-linear models. The main challenge of PFs is to prevent filter degeneracy through an appropriate resampling technique.

In our model, we use the particle Markov Chain Monte Carlo (MCMC) method (Andrieu et al., 2010; Šukys and Bacci, 2021), which is highly suitable for non-linear problems. The MCMC algorithm is used to infer selected hydrodynamic model parameters (see Sect. 2.4.2), with ensemble affine invariant sampler (EMCEE) for the parameter acceptance/rejection criterion.

For each proposed parameter set, the PF is used for data assimilation. A visualization of the process is shown in Fig. 2. To prevent filter collapse, we resample posterior trajectories according to their respective observational likelihoods before each sequential prediction. To the authors best knowledge, this is the first application of such a filtering algorithm to a fully three-dimensional model. A particular benefit of this approach is that the stochasticity from the atmosphere is sufficient to generate trajectories that manage to track the observational data with proper model parameters. This is in contrast to the

non-physical correction vector in many other DA schemes that are necessary to nudge trajectories toward the data. Aside from potentially causing instabilities, these approaches decrease the confidence in the fidelity of the underlying model, as the correction mechanism potentially also corrects a model deficiency.

### 2.4.2 Implementation using `SPUX` and design of numerical experiments

For data assimilation and particle filtering we use the `SPUX` package (Šukys and Bacci, 2021), a modular framework for parallel

Bayesian inference with a user-friendly programming interface. The 3D hydrodynamics package, `MITgcm`, was modified both to allow a Secchi depth value argument for every simulation hour and built as a shared library to enable interfacing with `SPUX` using the `ctypes` package. The `ctypes` approach, as opposed to launching the model as a subprocess, provided a noticeably faster and more stable performance. During calibration EMCEE was configured to run with 16 chains (distributed over 8 parallel workers), with 10 particles per filter; requiring a total of 89 parallel workers. The simulations were run on the

Swiss Supercomputing Center (CSCS) over a period of approximately three months ($\approx 10.5$ hours to predict 11 months using the PF, and $\approx 21$ hours for a full EMCEE sampler iteration).





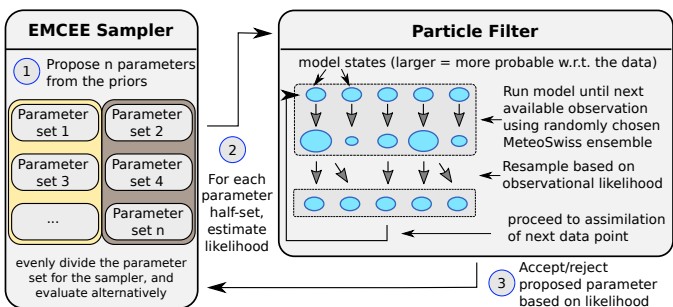

**Figure 2.** A visualization of the data assimilation framework. The EMCEE sampler proposes $n$ parameters, whose likelihood is then evaluated using the particle filter and used to determine whether to update that particular parameter.

As described in Sect. 2.2, we chose two hydrodynamic model parameters to infer. In both cases, we can establish an interval $(a, b)$ that contains the optimal value, but otherwise assume a uniform distribution, $U$, within. The first parameter is the Smagorinsky harmonic viscosity coefficient $C_{smag}$, to which we assign a prior distribution $U(2, 4)$, consistent with Griffies and Hallberg (2000). To the second parameter – the Dalton number, $C_D$ – we assign the prior $U(0.045, 0.06)$ based on a preliminary sensitivity analysis. For Dalton values outside the prior, heat exchanges with the atmosphere were not modeled satisfactorily. For example, the default `MITgcm` value $C_D = 0.0346$ results in significant temperature under-prediction during the summer months.

Observational data spanning January 15 - December 15, 2019 is used for the calibration and data assimilation (DA) run. Attempts to calibrate model parameters using a shorter timeframe generated posteriors that provided sub-optimal performance for the whole year (see supplementary material for more discussion), and therefore discarded. To analyze the effectiveness of model predictions, a control run (CR) is made without filtering and using parameters $C_D = 0.045$ and $C_{smag} = 2$ as the baseline.

### 2.4.3 Bulk-to-Skin conversion using LSTM

As the AVHRR operates in the infrared portion of the spectrum, it effectively measures the skin temperature in the top few millimeters of the lake. In the bulk region immediately below this surface layer, the temperature can be significantly different due to a number of factors such as wind and solar radiation (Wick et al., 1996; Alappattu et al., 2017). As the hydrodynamic model generates bulk temperature predictions, a bulk-to-skin (or skin-to-bulk) function is necessary. The available research, (e.g., Alappattu et al., 2017), generates estimates based on oceanographical studies. However, such approaches do not directly translate to lake research due to the differences in typical weather conditions, in particular, frequent low wind conditions over lakes (Bouffard and Wüest, 2019).

Thus we implement a Bi-directional Long Short-Term Memory (Bi-LSTM) neural network which uses a 27 hour history of 18 feature inputs to make a skin temperature prediction. 16 of the features come from the means and their respective spreads of the MeteoSwiss weather predictions (air temperature, cloud cover fraction, wind velocity, relative humidity, precipitation, short-wave and long-wave radiation). The last two features are the hydrodynamic model temperature predictions and hour of



day. The model was trained using data from 2018 and January 2020 - May 2021, with the bulk water temperature predictions extracted from the Meteolakes model (Baracchini et al., 2020b).

For the particle filter, uncertainty quantification of the predicted skin temperature is also necessary. Therefore, the Bi-LSTM model also implements additional methods to quantify epistemic and aleatoric uncertainty (Kendall and Gal, 2017). Monte Carlo dropout approximates predictions from an ensemble that can be used to quantify epistemic uncertainty. On the other hand,

using the negative log-likelihood of a normal distribution as the objective function in training allows Bi-LSTM to also estimate predicted variance that can be used to quantify aleatoric uncertainty. We generate 19 different skin temperature estimates and predicted variance for each input, from which we sum the average predicted variance and variance of the skin temperature estimates to obtain a scalar total variance. Accordingly, we construct a normal distribution with the computed total variance centered at the mean skin temperature prediction to be evaluated against the LSWT measurement.

## 3    Results and Discussion

In this section, we report on the inference results, with the initial focus on model parameter inference. As the calibration mechanism operates on distributions, not scalar quantities, the process is more complicated than what is typically done. Therefore this warrants a closer look at the posterior distributions and diagnostics. Then we compare the results obtained using the best posterior parameter set, and finally we evaluate the performance of the Bi-LSTM network as a predictor of skin temperature.

### 3.1    Hydrodynamic model calibration

The posterior distribution of two hydrodynamic model parameters (Smagorinsky viscosity and Dalton number) were estimated in SPUX using the EMCEE sampler (see Fig. 2). For each of the 16 parameter sets, the EMCEE sampler updates a parameter in case a better-performing one is found or it is deemed to be 'stuck' (no changes for 10 iterations). The PF is not guaranteed to choose the optimal global trajectory for a parameter set, given the computational constraints and the resetting of 'stuck' chains

(Šukys and Bacci, 2021), and therefore some uncertainty around the optimal parameter value is to be expected. In Fig. 3, we show the evolution of the Markov chain parameters in terms of their means and 5%-95% percentiles. We can observe from the relative stationarity of the distributions that the convergence to the true posterior distribution was likely achieved. We thus conclude that the optimal model parameters were localized with a sufficiently high degree of confidence.

The inferred marginal posterior distributions are shown in Fig. 4 in orange, with the prior distribution shown in blue. The

vertical red dashed lines indicate the best-found parameter set, with the values shown in the table to the right. In relation to their respective priors, the Dalton number is predicted with a relatively high degree of confidence, while the posterior for the Smagorinsky parameter is less defined. This is to be expected, as the Dalton number has a stronger effect on model predictions, and is therefore more sensitive. However, with more iterations, we expect that a more centered posterior for the Smagorinsky viscosity parameter would have been obtained.

We also consider the average redraw rate - the fraction of particles that form a basis for each sequential prediction - in Fig. 5. The survival rates dip significantly in cases when remote sensing data is assimilated or LéXPLORE data is available.



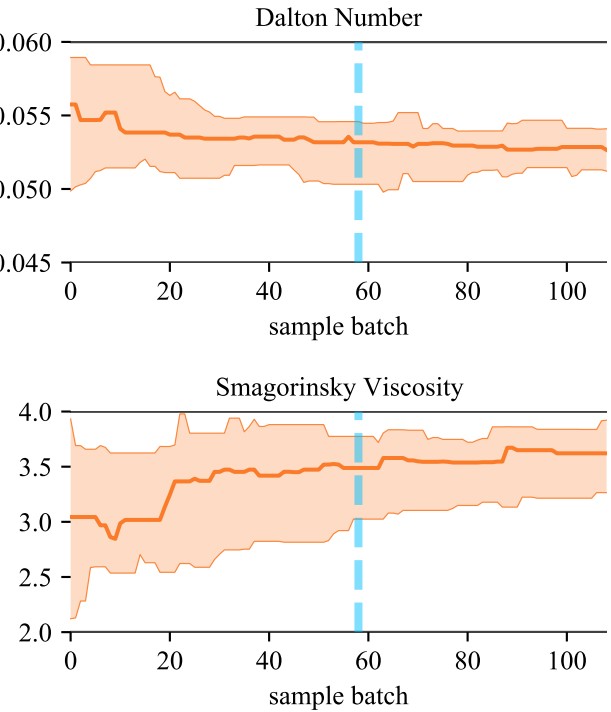

**Figure 3.** Evolution of the 16 Markov chain parameters. The solid lines indicate the median and the semi-transparent spreads indicate the 5% - 95% percentiles across multiple concurrent chains of the sampler. The vertical dashed blue line indicates the end of the specified burn-in period used to determine the posterior distribution.

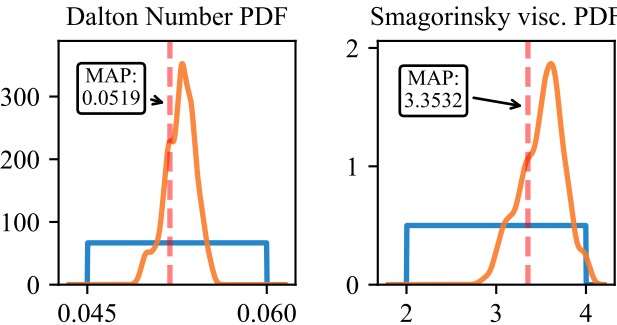

**Figure 4.** Marginal posterior (orange) and prior (blue) distributions of model parameters. The red dashed line indicates the maximum aposteriori parameter (MAP).

## 3.2 In situ data assimilation results

We present the results of the assimilated data predictions in this section, with the control run (CR) serving as the baseline for comparison. The data assimilation (DA) prediction was generated using the MAP values, given in Fig. 4 right. In addition,

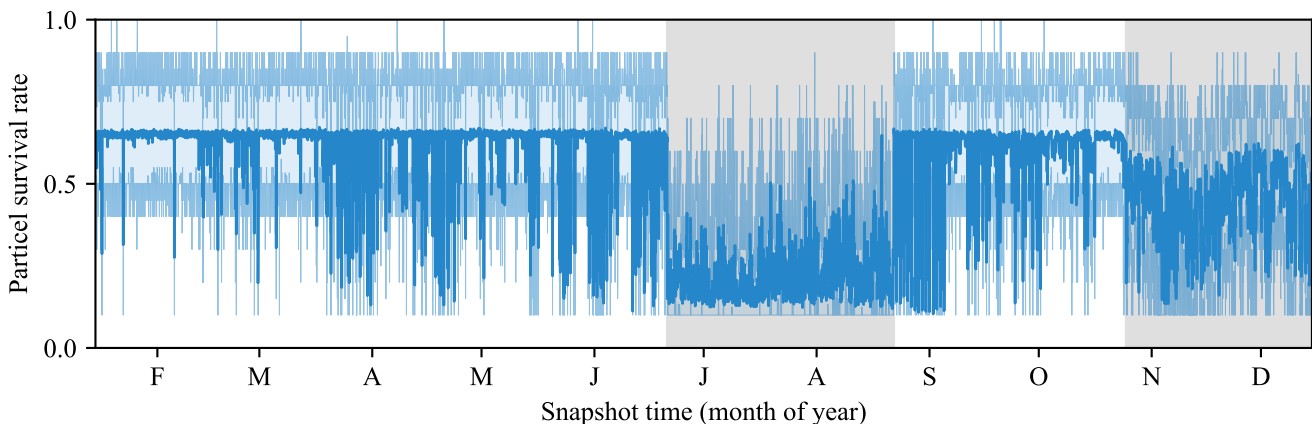

**Figure 5.** Particle survival rates (fraction of particles in the filter that are retained for continuing the DA) for each snapshot time in the PF likelihood estimator. The solid line indicates the mean, the semi-transparent lines indicate the 0%-100% percentiles. The periods of time when LéXPLORE data is available are shown as shaded gray regions.

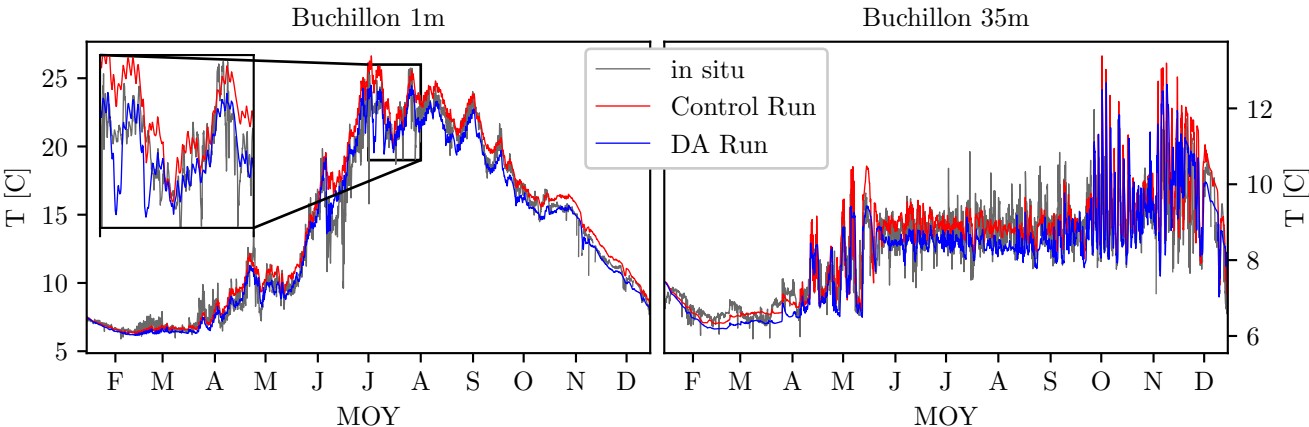

**Figure 6.** Evolution of temperature at the Buchillon station for the 1m (left) and 35m (right) sensors for the year 2019. The gray line is the in situ measurement, the red line corresponds to the control run, and the blue line is the data assimilated prediction. The small inset on the left shows the improvement from the assimilated run for the 1m sensor in greater detail for July.

whenever possible, we compare the aggregate error metrics to the values given in Baracchini et al. (2020b) for their 2017 data (specifically we do not use their 2019 data, which exhibits a noticeable drift in the deeper layers of the lake in comparison to SHL2 observations). The performance of the model for the Buchillon, SHL2 and LéXPLORE in situ datasets are considered in this section.

Figure 6 shows the temperature for the Buchillon station at depths of 1m (left) and 35m (right). The gray lines represent the
measurement, the red line shows the CR, and the blue line is the calibrated and data assimilated (DA) run. As evident from the results at both depths, the CR already captures the seasonal variation and the high-frequency fluctuations quite accurately. The





**Table 3.** Performance of the DA run across the different in situ datasets in comparison to the CR.

| Dataset | CR RMSE | CR MAE | DA RMSE | DA MAE |
|---|---|---|---|---|
| Buchillon [°C] | 0.95 | 0.61 | 0.77 | 0.54 |
| LéXPLORE Temp [°C] | 1.26 | 0.81 | 1.01 | 0.67 |
| LéXPLORE Vel [m/s] | 0.033 | 0.023 | 0.029 | 0.020 |
| SHL2 [°C] | 1.32 | 0.64 | 1.22 | 0.56 |
| GE3 [°C] | 1.45 | 0.84 | 1.25 | 0.76 |

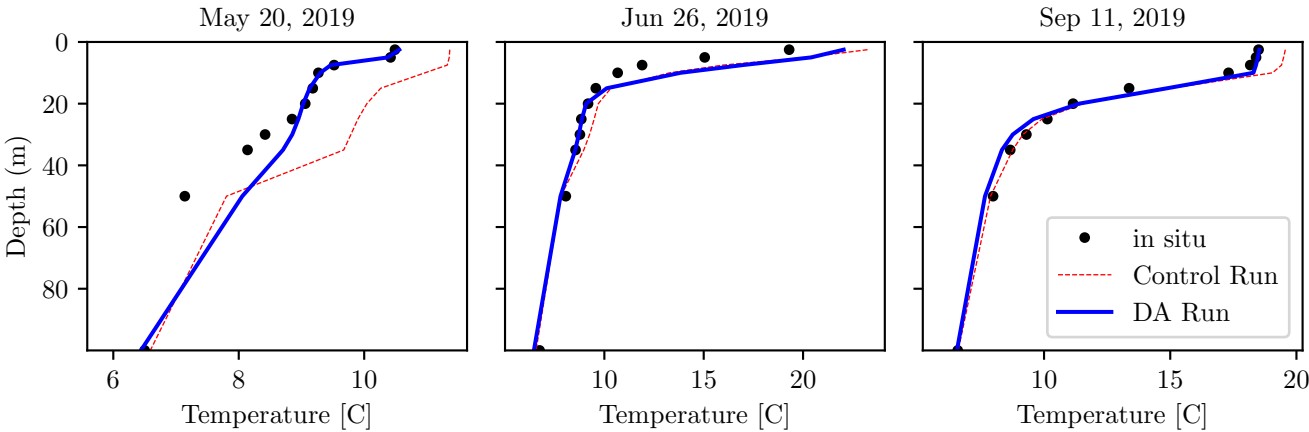

**Figure 7.** Vertical temperature profiles up to 100 meters deep at the SHL2 location. The black dots represent observations, the dashed red line is the control run, and the blue line shows the data assimilated run.

DA run improves model performance in the summer, where the CR under-predicts the near-surface temperature, as shown more clearly in the inset which focuses on the predictions for July. For the entire dataset, the RMSE decreases from 0.95 °C for the CR to 0.77 °C for the DA run. Overall, the improvement in RMSE and mean average error (MAE) across the various datasets

is 4-15%, summarized in Table 3. The main source of performance improvement is the better resolution of the near-surface temperature in summer.

We now consider the vertical temperature column profiles from the SHL2 location, which provides monthly measurements at the deepest location in the lake. As both the CR and DA run predictions below 100m are in agreement with the measurements, we focus on the upper portion of the column. In Fig. 7, we analyze the performance of the models for a few selected snapshots.

In the figure, the black dots represent measurements, the dashed red line is the CR, and the thick blue line is the DA run. The results show that the DA run tracks the observed temperature near the surface with greater accuracy, which also results in better modeling of deeper-layer temperature. Below 60 meters the observations are followed quite accurately by both the CR and DA run, without any substantial difference between the models.



Figure 8 compares the observed vertical column temperature at the LéXPLORE location to the DA run. The vast bulk of
measurements for this location (both temperature and velocity) were obtained during two periods, as reflected in the figure:
data for the period June 21 - August 11 is shown on the left, and the right side focuses on October 25 until December 15. For
simplicity, we will refer to the respective time intervals as 'summer' and 'autumn'. Outside of those time periods, the mea-
surements were sparse and therefore are omitted from comparison. The results show that the DA algorithm models the water
column temperature quite accurately, and in particular, correctly reproduces the thermocline depth in the summer. Furthermore,
the cooling cycle at the end of the year is captured quite well. Above the thermocline, the difference plots highlight that the
DA run over-predicts the temperature in the summer months, which as a result generates a smaller and dissipating warm bias
in autumn. The discrepancy could potentially be reconciled with a higher resolution model or more accurate Secchi depth
estimates. In general, correcting temperature over-prediction in the subsurface turbulent layer is a difficult problem exhibited
in many studies (Cimatoribus et al., 2018; Soulignac, Frédéric et al., 2018; Ye et al., 2020), without a clear consensus on the
underlying causes for each case.

Due to the coarse horizontal resolution of the mesh, the data assimilation algorithm primarily focused on tracking tem-
perature. As a result, the ADCP data played only a secondary role in determining the trajectory of the model. In Fig. 9, we
present the kinetic energy spectra computed from the LéXPLORE ADCPs (gray lines) and the DA run (blue lines) based on
different data sets. Figure 9 left, based on summer readings for the 15 meter sensor, shows excellent agreement for the kinetic
energy variability above the semi-diurnal (12h) mode. For the same sensor location, model predictions under-perform during
the autumn (Fig. 9 right), indicating that the high frequency internal wave modes are not being resolved by the hydrodynamic
model. A potentially large contributing factor for the discrepancies is the relatively coarse spatial resolution used in the model.

### 3.3 LSWT assimilation using BiLSTM network

On average, an LSWT image provided 209 usable pixels and significantly affected particle survival rates. For example, in
Fig. 5 most of the low survival rates are due to the AVHRR data, especially noticeable in cases when LéXPLORE data is not
available. In contrast to Baracchini et al. (2020b), where a highly selective criterion was applied, we use all the pixels which
have an associated non-zero QF value. This allowed a much more frequent remote sensing data assimilation, with 798 (of 2092
total) images usable for the data assimilation period. The use of LSWT has enhanced the model predictions by only a 4%
reduction of in situ RMSE. In Fig. 10, we show an example comparison between the observed LSWT (left), the hydrodynamic
model bulk temperature (center), and the BiLSTM prediction (right) for a selected snapshot. The result shows that BiLSTM
can predict the spatial variability and structure of LSWT images; although frequently it also generates an entirely different
profile. In general, as these improvement results are not particularly informative, we instead focus on the overall performance
of the assimilation model and the BiLSTM predictions.

The global training and performance of the BiLSTM model are summarized in Table 4. The training and testing used
the means and spreads of the MeteoSwiss weather predictions in combination with Meteolakes mean bulk temperature. The
results show that the network significantly improves predictions of LSWT for the training set, and achieves a 33% reduction
of RMSE for the test set. In the assimilated run, the bulk prediction difference is already small and only worse than BiLSTM



**Figure 8.** Vertical temperature profiles from the LéXPLORE sensors for summer months (left) and late autumn (right). The top row shows the dataset, the middle row shows the assimilated predictions. The bottom row shows the difference between the plots, with positive values indicating model over-prediction.

test set performance. In fact, the BiLSTM network increases the RMSE by about 10%, which most likely is attributable to the differences between the training data and the assimilation process. In general, as LSWT measurements carry significant uncertainty (1.3-1.5 °C RMSE), the analytic capability is limited by the lack of exact skin temperature measurements for 2019.

Surprisingly, the predictive capability of the BiLSTM seems to improve for the LSWT pixels with an associated QF in the range 2-5 (42% of the net pixel count), with an RMSE of 1.92 °C versus 2.11 °C for direct bulk comparison. For highest quality data (QF 6) however, BiLSTM does not improve performance. In Fig. 11 left, we analyze the performance of the BiLSTM (orange lines) against hydrodynamic model bulk predictions (in blue) for the different QFs by plotting mean RMSE with 10-90% percentiles. The gray bar chart shows the total number of LSWT measurements for the particular QF level. Aside from QF 5, the bulk RMSE gradually increases for lower QFs, as expected. At the same time, the BiLSTM error is practically constant with lower uncertainty, indicating the network's capability to predict lower fidelity data. For QF 5, the discrepancy



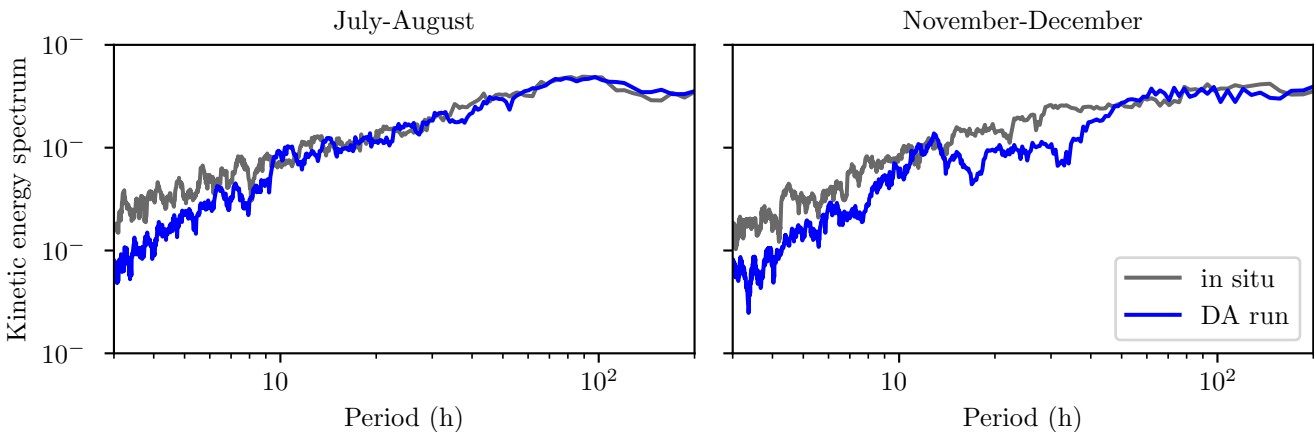

**Figure 9.** Kinetic energy spectra computed from the ADCP sensors at the LéXPLORE platform (gray) and from the DA run (blue). Profiles based on 15 meter deep dynamics use July-August data for the left figure and November-December data for the right figure.

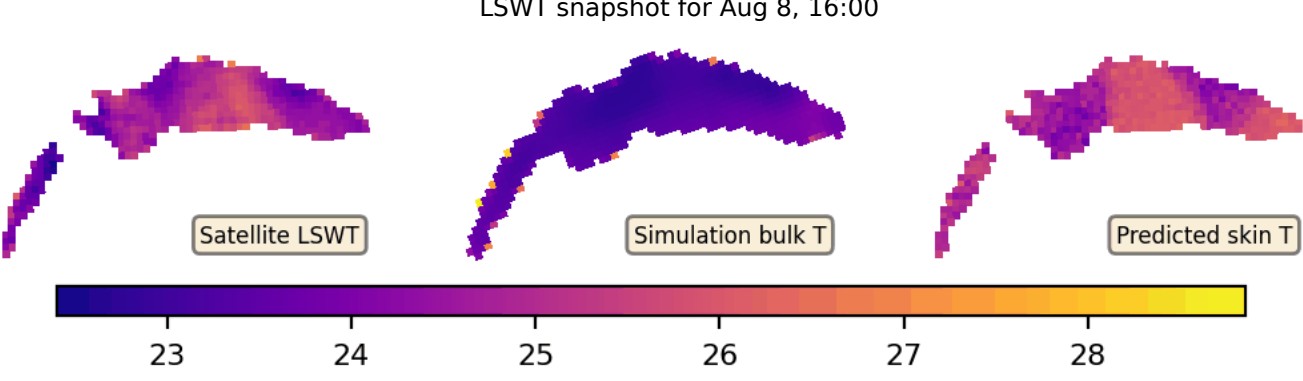

**Figure 10.** A comparison of LSWT snapshot from August 8, 2019 to the DA bulk and BiLSTM skin temperature predictions.

**Table 4.** Results of the BiLSTM model training (left two columns, with bulk temperature from Meteolakes (Baracchini et al., 2020b)), and performance in DA Run (right column).

|  | Training set (w. Meteolakes data) | Test set (w. Meteolakes data) | DA Run (SPUX-MITgcm) |
|---|---|---|---|
| Bulk RMSE | 3.00 | 2.37 | 1.85 |
| BiLSTM RMSE | 1.33 | 1.60 | 1.99 |





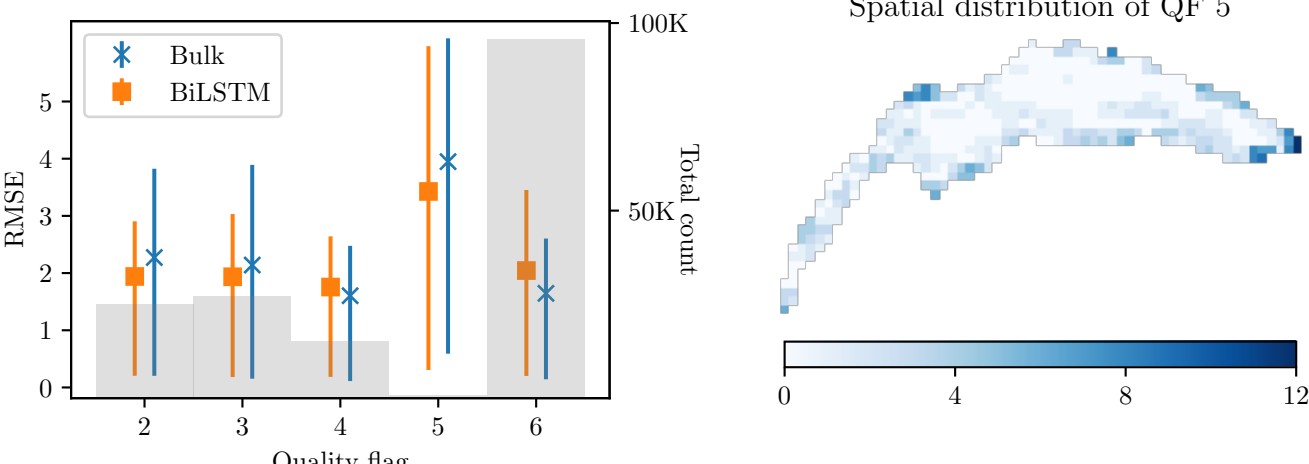

**Figure 11.** Analysis of Bi-LSTM performance in the DA run with respect to the different quality flags. Left: LSWT difference with respect to bulk temperature (in blue) and BiLSTM predictions (in orange) as a function of QF. The faint gray bars show the total number of LSWT measurements associated with the QF. Right: spatial distribution of LSWT measurement availability for QF 5.

nearly doubles, indicating a significant issue with this LSWT data subset. In contrast to the relatively uniform spatial frequency of other QFs on the lake, Fig. 11 right shows that most of the associated measurements occurred near the shore, which are
harder to predict accurately due to the resolution of the hydrodynamic model. However, as LSWT pixels with associated QF 5 are extremely rare (Fig. 11 left), they do not significantly contribute to the overall result.

To enable uncertainty quantification (UQ) for the PF, each individual bulk-to-skin prediction is also equipped with a normal uncertainty distribution, as described in Sect. 2.4.3. Since the network used hour of the day and weather prediction uncertainty as part of its training, we can expect some spatio-temporal variation in the predicted spreads. Therefore, we analyze those two
factors here. In Fig. 12 left, we show the hourly mean BiLSTM prediction difference with LSTM data as solid blue line, and the 10%-90% percentiles of the BiLSTM UQ as shaded blue regions. The results suggest that neural network UQ is capable of slightly better predicting the spreads for the different times of the day. In contrast, the default LSWT error model provides relatively uniform uncertainties across all times of the day, regardless of MAE (Fig. 12 right). In general, the improvement is however quite mild, potentially due to the low spatial resolution of the model.
Finally, we consider the spatial pattern of BiLSTM predictions, and in particular, examine whether its UQ can predict regions of the lake with larger model discrepancies. In Fig. 13 left, we present mean BiLSTM prediction RMSE for the different pixel locations over the lake. As can be expected, the best performance is obtained for off-shore pixels in the eastern portion of the lake (Grand Lac). The predictive capability of the network reduces nearshore, and especially in the western portion (in the Petit Lac). The BiLSTM uncertainty predictions follow a much similar pattern (Fig. 13 right), including the large increases in
uncertainty near the north shores. In part, this increase can most likely be attributed to the greater uncertainty in the atmospheric





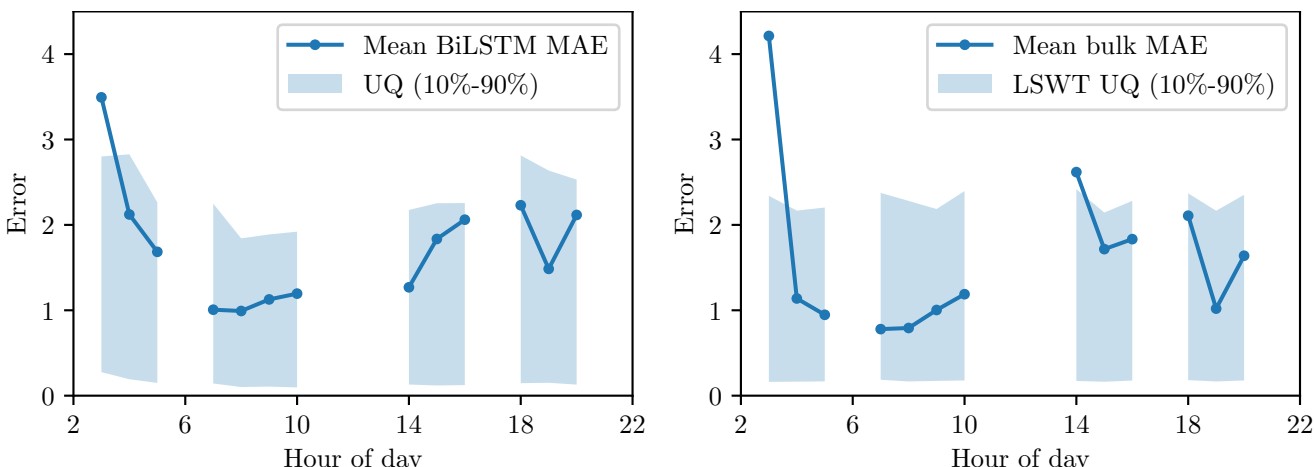

**Figure 12.** Left: mean BiLSTM prediction error (dark blue line) and the 10%-90% percentiles of the uncertainties predicted by Bi-LSTM (shaded areas) for the different hours of the day. Right: mean bulk temperature prediction error (dark blue line) and the 10%-90% uncertainty estimates based on Lieberherr and Wunderle (2018).

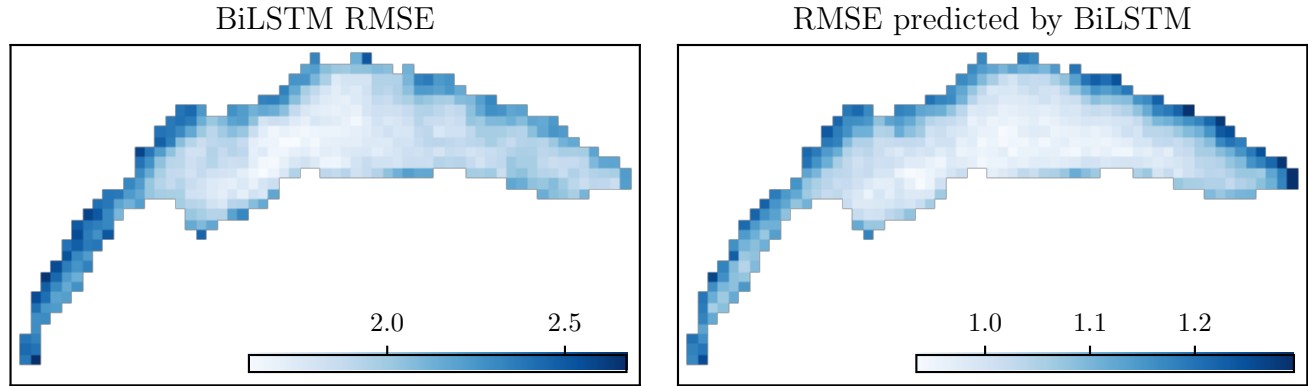

**Figure 13.** Spatial distribution for the BiLSTM RMSE (left) and the RMSE predicted by the BiLSTM.

weather conditions near the land-water interface, as well as the fact that the majority of observations for training BiLSTM come from off-shore in the Grand Lac, which would likely create a bias in the model predictions.

## 4 Conclusions

We presented the `SPUX-MITgcm` framework, a novel approach to the calibration of hydrodynamic model for highly spatio-

temporally heterogeneous observational dataset. The inference makes use of the ensemble affine invariant sampler (EMCEE) to infer the distribution of model parameters coupled with a particle filter (PF) for stochasticity in atmospheric forcing. The PF relied on resampling existing trajectories based on their observational likelihoods to infer the most probable weather conditions

over the lake. As a result, the PF generated physically realistic trajectories (at least, with respect to the hydrodynamic model). In addition, to enable the proper assimilation of remotely sensed lake surface temperature, we developed a Bi-directional
Long Short-Term Memory network for estimating lake skin temperature based on a history of weather and bulk temperature predictions.

The particle filter provides a relatively small improvement to model predictions (in contrast to other popular data assimilation schemes), but at no cost to the quality of the physical model. However, this approach requires a highly robust hydrodynamic model, as its corrective powers are limited. Despite the improvements, this approach is quite computationally costly, especially
as a tool for inferring model parameters. In addition, as discussed in the supplementary material, the sampler has significant difficulty with calibrating certain model parameters (although this issue could potentially be mitigated with a better error model). Therefore, we feel that a computationally cheaper method for parameter estimation (for example, an optimization algorithm instead of a sampler) might be the more productive approach. At the same time, an improved version of the particle filtering approach could provide a powerful option for operational forecasting models.

*Code and data availability.* The code used for these simulations, and an example portion of the data, are openly available. The SPUX source code is available at https://doi.org/10.5281/zenodo.5638313, the modified MITgcm repository can be found at https://doi.org/10.5281/zenodo.5634042, and the repository for handling MITgcm runs is at https://doi.org/10.5281/zenodo.5637216. While we cannot openly publish the forcing data, a representative snapshot is available at https://renkulab.io/gitlab/artur.safin/datalakes-observational-data-snapshot, and the 3D mean temperature and velocity predictions can be accessed at https://doi.org/10.5281/zenodo.5642898. Finally, the reproducibility
of the computational environment is enabled through RENKU (Swiss Data Science Center, 2021), an online platform for the storage, tracking and replication of numerical codes, which enables users to launch the container directly from the RENKU site. The entire SPUX-MITgcm repository and the computational environment is available as a docker at https://renkulab.io/projects/artur.safin/DatalakesHydrodynamics, and an example of running the inference is in the supplementary material.

*Author contributions.* JŜ and DB designed the research, AS implemented the data assimilation framework and ran the inference. Šukys
extended SPUX package to be compatible with MITgcm. FO implemented and trained the BiLSTM network. DB and CLR advised with the hydrodynamic model. All authors assisted with discussions, dataset management and manuscript preparation.

*Competing interests.* The authors declare that they have no conflict of interest.

*Acknowledgements.* This work was funded by the Swiss Data Science Center (SDSC Project DATALAKES C17-17) and Eawag Discretionary Funding. We would like to thank the entire team from LéXPLORE platform, for their administrative and technical support and for
LéXPLORE core dataset. We also acknowledge LéXPLORE five partner institutions: Eawag, EPFL, University of Geneva, University of



Lausanne and CARRTEL (INRAE-USMB). For SHL2 data, we acknowledge the Observatory of alpine LAkes (OLA), SOERE OLA-IS, AnaEE-France, INRA of Thonon-les-Bains and CIPEL. We would like to thank the Swiss Federal Office for the Environment (BAFU) for river and water level data and Stefan Wunderle and the Remote Sensing group at the University of Bern for LSWT data. The authors would also like to thank Marco Bacci, Theo Baracchini, Fernando Perez-Cruz, Eric Bouillet and Daniel Odermatt for helpful discussions.





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
