# Peer review of "A Bayesian data assimilation framework for lake 3D hydrodynamic models with a physics-preserving particle filtering method using SPUX-MITgcm v1"

_Geoscientific Model Development, 2021_

## Author Comment (AC1)

**Response to RC1**

We thank the anonymous reviewer for their reading and suggestions. We have made some changes in the manuscript to reflect the concerns, and address them below, with the referee comments in blue.

*Comment:*

General comments: This paper introduces a noval DA approach for lake hydrodynamic model predictions. It is well motivated and very innovative piece of work, but I do find some major issues with the clarity in description of the new method, which make the results a bit hard to understand. The design of the DA approach is quite complicated, combining several non-traditional methods, and the application scenario is also quite different from the typical initial-value prediction problems. I suggest the authors to improve the paper with clearer presentation of each component (particle filter, neural network, and sampler) and discuss how each of them contributes (compares to traditional methods) to improve the accuracy of prediction. This would help convince the readers that such a novel approach has potential for further applications.

*Answer:*
We have reworked the description of the methodology with additional details about the sampler and the particle filter, and enhanced Figure 2 to provide a comprehensive visualization of the data assimilation framework and the particle filter, and an explanation of the error model where the Bi-LSTM network is used. The paragraph describing the EMCEE sampler now reads:

*In our model, we employ a particle Markov Chain Monte Carlo (MCMC) method which is highly suitable for non-linear problems (Andrieu et al., 2010; Šukys and Bacci, 2021). The MCMC algorithm is used to infer selected hydrodynamic model parameters (see Sect. 2.4.2), with ensemble affine invariant sampler (EMCEE) for the parameter acceptance/rejection criterion. A visualization of the process is shown in Fig. 2. The EMCEE sampler is particularly effective for poorly scaled distributions that become well-conditioned under affine transformations, and can be significantly faster than standard MCMC approaches on highly skewed distributions. A more thorough discussion of the advantages and disadvantages is given in Goodman and Weare (2010). Here, we only provide a brief overview of the mechanism. The EMCEE algorithm initializes with an ensemble of Markov chains (walkers), $\{X_i\}$, drawn from a prior probability distribution $\Pi(x)$, and split into two subsets, $S_1$ and $S_2$, and their marginal likelihood is estimated as explained in the paragraph below. First, we update all the walkers $X_j$ from $S_1$ using the stretch formula $X_{j,new} = X_j + Z[X_j - X_k]$, where $X_k$ is a randomly chosen walker from $S_2$, and Z is a scaling variable. Each proposed update $X_{j,new}$ is confirmed in accordance with a Metropolis acceptance probability. The next step is to update $S_2$ walkers using the updated $S_1$ elements. We continue alternatively evolving walkers from $S_1$ and $S_2$ until a suitable convergence criterion is met.*

With regards to the particle filter, we now have:

*For each EMCEE parameter, a particle filter (PF) is deployed to address the stochasticity of the weather predictions. The PF, implemented in Šukys and Bacci (2021), works as follows. For each parameter $\alpha^{(k)}_i$, we initialize m model states, $M_j$, and simulate until an observation is reached at time $t = t_p$ (see Fig. 2). At this point, model simulations are paused and all particles are resampled (bootstrapped) according to their observational likelihoods. Thus, certain model states will be deleted and replaced by another state from a different trajectory. Such a resampling algorithm significantly increases algorithmic and implementation complexity due to the required destruction and replication of existing particles. However, it also provides an efficient way of*

*sampling "intermediate" posterior model states. To the authors best knowledge, this is the first application of such a filtering algorithm to a fully three-dimensional model. A particular benefit of this approach is that the stochasticity from the atmosphere is sufficient to generate trajectories that manage to track the observational data with proper model parameters. This is in contrast to the non-physical correction vector in many other DA schemes that are necessary to nudge trajectories toward the data. Aside from potentially causing instabilities, these latter approaches decrease the confidence in the fidelity of the underlying model, as the correction mechanism potentially also corrects a model deficiency. At the same time, as no model is perfect, the SPUX PF offers limited capability to handle biases that the sampler does not eliminate. In addition, the strict nature of our PF (model states cannot be modified!) significantly limits its performance, and thus it cannot expect to outperform the above-mentioned established alternatives.*

For the BiLSTM neural network, we significantly expanded the description of the framework, and also added Figure 3, which shows a schematic of how the BiLSTM network operates. We also added a reference to another work by Stadler et al. (2021, https://arxiv.org/abs/2109.13235v1), which provides an extension the framework to graph neural networks to account for the inherently spatial nature of the LSWT data.

*Comment:*

The authors claim that this paper serves as a proof of concept for particle filtering in other higher-dimensinoal problems (Lines 44-47). This is misleading since the particle filter component only updates (infers) two parameters from the hydrodynamic model, and the larger-dimensional model states are updated through the BiLSTM network. You are effectively running the PF in a reduced-dimension system, using a nonlinear operator to map between the full model states and the reduced states. So this needs to be clarified.

*Answer:*

We made some changes in the manuscript to clarify the role of the BiLSTM network. The purpose of the BiLSTM network is only to generate an estimate of the lake skin temperature from the hydrodynamic bulk temperature predictions. The bulk-to-skin (or skin-to-bulk) conversion is a necessary component of data assimilation, as the hydrodynamic model only produces bulk temperature, and the difference between bulk and skin temperature can be significant, especially in the summer. The BiLSTM network, however, is only used in the error model to help estimate the marginal likelihood of the trajectory (see Fig. 2 in the text) and does not alter the physical state of the model. Rather, it helps the PF select the best-performing trajectories in the particle replication/deletion stage.

We also would like to point out that our data assimilation framework can be run without the BiLSTM network – either through the omission of LSWT data, or by using a simpler error and uncertainty model from Lieberherr and Wunderle (2018), which is less accurate (see Section 2.3.2).

*Comment:*

Related to #1, how challenging is the lake model prediction problem compared to for example mesoscale weather prediction? The difficulty in weather prediction is the chaotic nature of convections that amplifying initial condition errors rapidly. You mentioned that lake dynamics are also quite volatile (Line 126) and small errors can impact the model trajectory. However, you chose to estimate model parameters, which seems to be related to atmospheric boundary forcings, instead of the initial condition errors in lake states. Does this mean the problem is more in boundary forcing rather than initial conditions? Estimating model parameters are quite different from estimating the states, so this needsto be defined clearly.

*Answer:*

Initial conditions are quite a difficult problem, as in-situ observations are limited (typically 1-2 array of sensors at best). In our case, only the measurements at SHL2 provided an entire single-point vertical temperature profile of the lake, which is necessary to generate an initial condition for the model, under the assumption of homogeneous lateral temperature and zero velocity. As we intentionally initialize the model in winter, fluctuations in water temperature are minimal during that period, and thus such initial conditions are reasonably accurate; the model "spins up" in a reasonably short amount of time (1 month or so for velocity to spread from surface to the deeper layers).

The most volatile portion of the lake is near the surface, and surface coupling is a highly crucial component, as energy exchanges and atmospheric conditions provide the dominant source of dynamics that eventually propagate into the deeper layers of the lake. Thus, correct interface parameters are essential. At the same time, a large source of stochastic uncertainty comes from the uncertainty in weather predictions, and this is where particle filtering can help. So the problem of determining the correct boundary forcing conditions is indeed essential.

While the last statement might be well-suited for atmospheric sciences, in limnology the process is different. In our view, estimation of model parameters is inherently coupled to state estimation, at least in limnology. States are a necessary means of determining model error and uncertainty and therefore provide feedback on the choice of the model parameter. The primary focus is to estimate the model parameters, but at the same time we can also extract model predictions associated with the best-performing parameter set.

*Comment:*

The introduction of BiLSTM in section 2.4.3 could be improved if you adopt standard terminology in DA. For example, the bulk-to-skin conversion is essentially the observation operator, or forward operator, that maps lake model state variables (state space) to the observed skin temperature (observation space). A discussion of why using a neural network for this nonlinear function, rather than using some physical model, may help the reader understand better.

*Answer:*

We added the suggested terminology (forward operator, observation space) to section 2.4.3. We also added a discussion of what a bulk-to-skin parametrization would require. The relevant portion now reads:

*As the hydrodynamic model generates bulk temperature predictions, a bulk-to-skin function (forward operator) is necessary for the observation space error model. Existing estimates based on oceanographical studies - (e.g., Alappattu et al., 2017) - do not directly translate to lake research due to the differences in typical weather conditions such as frequent low wind conditions over lakes (Bouffard and Wüest, 2019). For lakes, an accurate bulk-to-skin parametrization would incorporate effects due to convectively driven surface turbulence (Wilson et al., 2013), and more accurate implementation of the air-water gas exchange, especially in the presence of surfactants (Bouffard and Wüest, 2019). While such a parametrization might be possible, it would be technically difficult to implement, and therefore we attempt a different approach.*

Ultimately, the implementation and testing of such a parametrization would require quite a bit of additional time and code modification, and there is no guarantee that it would perform well. Therefore, in the interest of avoiding feature creep, we decided to implement the BiLSTM network.

*Comment:*

> It is still a bit unclear what exactly are being estimated, the two parameters or the whole model states, in the Bayesian framework described in section 2. Figure 2 only shows the updating of the two parameters using the particle filter and sampler, but how does this connect to the observation (skin temperature) and other model states (temperature profiles)? Does the updated parameters change model states through a nonlinear model run? Maybe extending the schematic diagram to include all components and clarify their connection would help.

*Answer:*

The primary focus of the paper is the Bayesian framework to calibrate hydrodynamic model parameters *in terms of distributions*. At the same time, however, analysis of posterior model states is essential to understand the effectiveness of the calibration process and the particle filter. Figure 2 has been reworked to show a more comprehensive picture of the framework, and now has the error model as well.

The parameters remain constant within a single particle filter evaluation. The conclusion of the filtering process, however, returns the marginal likelihood of the parameter, which is used by the EMCEE sampler in the next parameter update step.

*Comment:*

> Line 25: the fact that EnKF only assimilated a fraction of LSWT data is surprising, could you explain more what is the limitation? Is it because the high spatial heterogeneity that cause nonlinearity?

*Answer:*

There were two primary reasons to limit the frequency of assimilation of LSWT data:

1) LSWT measurements, due to coming from the satellite, can have significant deviations from the truth due to a number of issues, such as zenith angle, atmospheric conditions and sensor defects (see Section 2.3.2 for references and additional discussion). Therefore, the author of the MeteoLakes EnKF model decided to only assimilate at times when there was high confidence in the LSWT measurement data (so only when the associated quality factors were high, and a sufficiently large number of pixels over the lake were available).

2) In addition, the potentially large difference between the bulk and skin temperatures had to be taken into account. As the MeteoLakes EnKF model did not incorporate a bulk-to-skin operator, they could only use LSWT data under the right conditions, namely when the top layer was sufficiently well mixed, making the bulk-skin difference sufficiently small.

*Comment:*

> Line 30: particle method: do you mean particle filter method?
> Line 145: add reference for EnKF (Evensen 1994), "highly popular blend", do you mean "brand"?
> Line 154: add references for particle filter, and filter degeneracy issue and resampling technique

*Answer:*

Corrected, as suggested. The particle filter and the resampling technique are described in Šukys and Bacci (2021), which has been added in the text. We removed the reference to particle degeneracy, but a good discussion on the problem is the article "A Tutorial on Particle filtering and smoothing: Fifteen years later" by Doucet and Johansen (2009).

*Comment:*
    Line 159: what does EMCEE stands for? could you add a reference for this sampler?

*Answer:*
EMCEE is the ensemble affine invariant sampler, first proposed by Goodman and Weare (2010). An additional reference to the publication has been added in Section 2.4.1, *Choice of DA model*.

*Comment:*
    Line 174: 10 particles per filter, ... 89 parallel workers. Figure 2 states n sets of parameters, so n=10 here? Please clarify.

Answer:
SPUX implementation of the EMCEE sampler that we use requires additional parallel workers just to manage the chains (for speed and stability reasons). For the 16 chains that we use, only 8 can be run in parallel as per EMCEE rules. For each chain, we assign 10 particles, which requires 80 total workers. In addition, each chain manager runs on a dedicated process (+8 workers) and the global inference manager also requires its own process (+1 worker). Thus, we have 80 + 8 + 1 = 89 workers. The 2n in Figure 2 refers to the number of chains present in the sampler. We have added a clarification in Figure 2 that connects the 2n value to the number of chains in the SPUX sampler. We have clarified this point in the manuscript as well (line 196).

Comment:
    Line 179: how is the uniform distribution and the upper/lower bounds chosen? Based on physical intuition or some prior studies?

For the Dalton number, the boundaries were chosen on the basis of a few preliminary deterministic runs, which indicated that the values outside the priors would provide poor performance. For Smagorinsky viscosity, the boundaries were chosen based on physical intuition, and also confirmed by a few more preliminary runs.
   In general, you could choose wider boundaries for the prior distributions, and you would still obtain convergence with a slightly longer run.

---

## Author Comment (AC2)

**Response to RC2**

We thank the anonymous reviewer for their reading and suggestions. We have made some changes in the manuscript to reflect the concerns, and address them below, with the referee comments in blue.

*Comment:*

This paper presented the SPUX-MITgcm framework, an approach to the calibration of a hydrodynamic model for a highly spatiotemporally heterogeneous observational dataset. The current form of the paper contains a lot of information and model exercises but was not presented in the most suitable way (missing key information or details ) to guide the readers to well understand the value of the work. As such there are some difficulties in evaluating the work.

*Answer:*
We have reworked the description of the methodology with additional details about the sampler and the particle filter, and enhanced Figure 2 to provide a comprehensive visualization of the data assimilation framework and the particle filter, and an explanation of the error model where the Bi-LSTM network is used. The paragraph describing the EMCEE sampler now reads:

*In our model, we employ a particle Markov Chain Monte Carlo (MCMC) method which is highly suitable for non-linear problems (Andrieu et al., 2010; Šukys and Bacci, 2021). The MCMC algorithm is used to infer selected hydrodynamic model parameters (see Sect. 2.4.2), with ensemble affine invariant sampler (EMCEE) for the parameter acceptance/rejection criterion. A visualization of the process is shown in Fig. 2. The EMCEE sampler is particularly effective for poorly scaled distributions that become well-conditioned under affine transformations, and can be significantly faster than standard MCMC approaches on highly skewed distributions. A more thorough discussion of the advantages and disadvantages is given in Goodman and Weare (2010). Here, we only provide a brief overview of the mechanism. The EMCEE algorithm initializes with an ensemble of Markov chains (walkers), $\{X_i\}$, drawn from a prior probability distribution $\Pi(x)$, and split into two subsets, $S_1$ and $S_2$, and their marginal likelihood is estimated as explained in the paragraph below. First, we update all the walkers from $S_1$ using the stretch formula $X_{j,new} = X_j + Z[X_j - X_k]$, where $X_k$ is a randomly chosen walker from $S_2$, and $Z$ is a scaling variable. Each proposed update $X_{j,new}$ is confirmed in accordance with a Metropolis acceptance probability. The next step is to update $S_2$ walkers using the updated $S_1$ elements. We continue alternatively evolving walkers from $S_1$ and $S_2$ until a suitable convergence criterion is met.*

With regards to the particle filter, we now have:

*For each EMCEE parameter, a particle filter (PF) is deployed to address the stochasticity of the weather predictions. The PF, implemented in Šukys and Bacci (2021), works as follows. For each parameter $\alpha^{(k)}_i$, we initialize m model states, $M_j$, and simulate until an observation is reached at time $t = t_p$ (see Fig. 2). At this point, model simulations are paused and all particles are resampled (bootstrapped) according to their observational likelihoods. Thus, certain model states will be deleted and replaced by another state from a different trajectory. Such a resampling algorithm significantly increases algorithmic and implementation complexity due to the required destruction and replication of existing particles. However, it also provides an efficient way of sampling "intermediate" posterior model states. To the authors best knowledge, this is the first application of such a filtering algorithm to a fully three-dimensional model. A particular benefit of this approach is that the stochasticity from the atmosphere is sufficient to generate trajectories that manage to track the observational data with proper model parameters. This is in contrast to*

*the non-physical correction vector in many other DA schemes that are necessary to nudge trajectories toward the data. Aside from potentially causing instabilities, these latter approaches decrease the confidence in the fidelity of the underlying model, as the correction mechanism potentially also corrects a model deficiency. At the same time, as no model is perfect, the SPUX PF offers limited capability to handle biases that the sampler does not eliminate. In addition, the strict nature of our PF (model states cannot be modified!) significantly limits its performance, and thus it cannot expect to outperform the above-mentioned established alternatives.*

Thus, the difference is largely in the different design goals of the particle filtering methods. The SPUX PF seeks a realistic trajectory (at least within the confines of the hydrodynamic model), while EnKF and 4-Var seek to minimize model prediction errors.

*Comment:*
> Details how the PF is configured and why, sensitivity analysis, convergence rate, etc.

We added a reference to the source material (*Šukys and Bacci, 2021*) where the PF is described in detail. In this paper, we provided all the main points of the filtering algorithm, and provide the relevant configuration details in Section 2.4.2. The number of particles/chains was largely determined by the size of the computational allocation at the CSCS cluster, and the amount of time the inference would require. We do not find these considerations particularly relevant, and thus do not include them in the text.

*Comment:*
> The same for LSTM, LSTM framework should be introduced, how does it work, describe
> training and validation process and evaluation.

We modified the text to provide a more thorough description of the BiLSTM network we use in the model, together with a description of the data that we used to train it in Section 2.4.3. We also added Fig. 3, which provides a visualization of the operational BiLSTM module that we use. We also added a reference to a paper that provides an extension of the framework to accoutn for spatial features (Stalder et al., 2021). The relevant portion of Section 2.4.3 now reads:

> *[…] We implement a neural network using Bi-directional Long Short-Term Memory (BiLSTM) blocks that use the 27-hour history of 18 feature inputs to make a skin temperature prediction. 16 of the features come from the means and their respective spreads of the MeteoSwiss weather predictions (air temperature, cloud cover fraction, wind velocity, relative humidity, precipitation, short-wave and long-wave radiation). The last two features are the hydrodynamic model temperature predictions and hour of the day. The model was trained using data from 2018 and 2020, with the bulk water temperature predictions extracted from the Meteolakes model (Baracchini et al., 2020b). Data from 2019 was separated from the training for benchmarking purposes. A schematic of a BiLSTM block and the neural network is shown in Fig. 3. Input to the neural network is provided as a (27 time-step, 18 channel) tensor. An initial fully connected layer maps these 18 channels to 32 channels. The output is then sequentially passed through three separate BiLSTM cell blocks. Finally, the output of the last BiLSTM block is linearly mapped to a two-channel output, which corresponds to temporal predictions of the skin temperature and their predictive log-variances. In our experiments, using LSTMs were crucial in order to exploit historical patterns in the input features when predicting the skin temperature. In a recent work, an extension of the LSTMs to a spatially-dependent model is presented by Stalder et al. (2021).*

*For the particle filter, uncertainty quantification of the predicted skin temperature is also necessary. Accordingly, the BiLSTM blocks in our neural network randomly disables 30% of the LSTM units to induce stochasticity. This allows our model to also implement additional methods to quantify epistemic and aleatoric uncertainty (Kendall and Gal, 2017). Monte Carlo dropout approximates predictions from an ensemble that can be used to quantify epistemic uncertainty. On the other hand, using the negative log-likelihood of a normal distribution as the objective function in training allows BiLSTM to also estimate predicted variance that can be used to quantify aleatoric uncertainty. Specifically, for a given input of 18 feature observations over a 27-hour history, the neural network predicts the corresponding 27-hour skin temperature estimations as well as their estimated logarithmic variances. During the optimization phase of the neural network, negative log-likelihood of a normal distribution as in Kendall and Gal (2017) (eqn 8), is used with skin temperature estimations ($\hat{y}_i$) and corresponding logarithmic variance estimation ($s_i$). At test time, we generate model predictions for 19 times for each input while keeping dropouts within BiLSTMs active, yielding 19 different prediction vectors, similar to an ensemble model. While the mean of estimated skin surface temperature are used for mean estimates, we use the variance of the 19 predictions for epistemic uncertainty. Aleatoric uncertainty is computed from the predicted variance estimates by taking their average after mapping the predicted logarithmic variance into variance. The total scalar variance is computed as the sum of the two variances. Accordingly, we construct a normal distribution with the computed total variance centered at the mean skin temperature prediction to be evaluated against the LSWT measurement.*

*Comment:*
how does EMCEE Sampler work, how did you evaluate its performance?

Section 2.4.1 has been rewritten to provide greater detail on the EMCEE sampler, and Figure 2 has been greatly expanded to visualize the process of sampler initialization, parameter proposal via the "stretch" move, and an example of how the parameters might evolve.

The performance of the EMCEE sampler is evaluated in Section 3.1, where we conclude that the sampler has converged based on the evolution of the Markov chain parameters, and therefore the sampler has performed satisfactorily. On the other hand, in the supplementary material Section 3 we demonstrate a case of the sampler diverging in an attempt to calibrate dynamic model parameters (eddy diffusivity/viscosity). Thus, the sampler is not guaranteed to work. In the conclusion, we state:

*In addition, as discussed in the supplementary material, the sampler has significant difficulty with calibrating certain model parameters (although this issue could potentially be mitigated with a better error model). Therefore, we feel that a computationally cheaper method for parameter estimation (for example, an optimization algorithm instead of a sampler) might be the more productive approach. At the same time, an improved version of the particle filtering approach could provide a powerful option for operational forecasting models.*

In the end, we conclude is that the approach will require more work to be viable for 3D data assimilation problems.

*Comment:*

I am also not quite sure about the BiLSTM's role in the proposed framework.

We improved the presentation of the methodology to be clearer as to how the BiLSTM network fits into the framework. As shown in the revised Figure 2, the purpose of the network is to provide an observation operator, which maps the predicted bulk temperature from the hydrodynamic model onto skin temperature to be compared to LSWT data.

Comment:

A better highlighting of the novelty and achievement of the work in the context with comparison to a similar or alternative approach. My impression is that it is a novel framework applied to the 3-D model, but not fully sure how effective and efficient it improve the simulation results. I think this can be significantly improved if the authors can re-structure the manuscript to highlight key information about the models used in the study.

In terms of parameter calibration, we agree that a comparison to a different MCMC type of sampler or a completely different calibration methodology (such as DUD) could be quite beneficial. However, such a comparison would require the implementation of such a sampler in SPUX which would also be compatible with the hydrodynamic model, MITgcm. This would require a significant time investment, and thus was not completed. At the end of the paper, we conclude that further developments, such as a more effective PF, are necessary before the framework becomes usable.

---

## Author Response (AR2)

**Response to Reviewer Comments**

We thank the anonymous reviewer for their reading and suggestions. We have made some changes in the manuscript to reflect the concerns, and address them below, with the comments in blue.

*Comment:*

It is known that particle filter handles nonlinearity well in low-dimensional systems, but it has some limitation when dimensionality scales up. The work presented used a clever separation between the model parameters (reduced dimension space) and actual high-dimensional model states, so that particle filtering happens in the reduced space so you don't worry about filter degeneracy. To apply to the high-dimensional lake simulation problem, clearly some other components (the sampler, the nonlinear BiLSTM) are needed to go back and forth between the reduced-dimension space and the full physical space. However, in your introduction Line 45-48 the DA and particle filtering approach is motivated as a proof of concept for "other higher-dimensional problems", which sounds too ambitious to me. While you successfully demonstrated the used of particle filtering for model parameters and sampling of model states in this particular lake model case, the method is not general to all high-dimensional problems. Is it possible to always come up with a reduced dimensional space (some model parameters are not global) so that this approach can be applied? I hope the authors can consider rephrasing this motivation in the introduction, maybe making it more clear that the idea is to apply particle filtering in a low-dimensional space to ensure its performance, so that readers are not misled.

We agree that the statement in the final introduction paragraph is a bit too ambitious, at least in comparison with the results shown in this paper. The new version provides a more modest statement:

*We investigate the viability of this approach and analyze the performance of individual components. The results demonstrate that while our methodology improves model performance, the framework requires further improvements to become usable for practical applications.*

At the same time, we make a note that the Bayesian inference package we used (SPUX) is capable of inferring model parameter distributions in relatively high-dimensional parameter spaces (10 or more parameters [Sukys, J. 2020, unpublished], or several time-dependent parameters [Bacci et al 2022]), but with significantly less computationally demanding state models. Typically hydrodynamic lake models do not stipulate many parameters to be calibrated, and a good fidelity model can be obtained with the calibration of around 5 appropriately chosen parameters. With a better and more sensitive error model together with a shorter dataset (~1 month of observation instead of the original 11 months), we believe that we would have been able to achieve good results in a higher-dimensional parameter space.